# Post-Mortem Computed Tomography Pulmonary Findings in Harbor Porpoises (*Phocoena phocoena*)

**DOI:** 10.3390/ani12111454

**Published:** 2022-06-04

**Authors:** Nienke W. Kuijpers, Linde van Schalkwijk, Lonneke L. IJsseldijk, Dorien S. Willems, Stefanie Veraa

**Affiliations:** 1Department of Clinical Sciences, Division of Integrating Disciplines, Faculty of Veterinary Medicine, Utrecht University, Yalelaan 108, 3584 CM Utrecht, The Netherlands; d.s.willems@uu.nl (D.S.W.); s.veraa@uu.nl (S.V.); 2Department of Biomolecular Health Sciences, Division of Pathology, Faculty of Veterinary Medicine, Utrecht University, Yalelaan 1, 3584 CL Utrecht, The Netherlands; l.vanschalkwijk@uu.nl (L.v.S.); l.l.ijsseldijk@uu.nl (L.L.I.)

**Keywords:** cetacean, virtopsy, necropsy, pulmonary pathology, decomposition

## Abstract

**Simple Summary:**

The use of cross-sectional imaging techniques to examine the cause of death and health status of deceased animals is increasing in both veterinary and wildlife conservation programs, including species of whales and dolphins. Lung disease is common in harbor porpoises (*Phocoena phocoena*), a small whale species that regularly washes up on the coast in North Sea-bordering countries. This study aimed to describe lung changes visible in computed tomographic (CT) images of recently deceased harbor porpoises before pathological dissection was performed, including comparison of these two examination methods. Despite frequently visible signs of body decomposition, several lung abnormalities (collapsed lung, fluid in the airways, lung mineralization) were more often seen on the CT images. In general, lung changes could be described in more detail compared to gross dissection. CT images of lungs of recently deceased harbor porpoises can therefore be used to guide gross dissection, leading to more specific findings and potentially a more complete understanding of the circumstances leading to the death of the porpoise, assessment of the population, and ultimately, ecosystem health.

**Abstract:**

The application of whole-body post-mortem computed tomography (PMCT) in veterinary and wildlife post-mortem research programs is advancing. A high incidence of pulmonary pathology is reported in the harbor porpoise (*Phocoena phocoena*). In this study, the value of PMCT focused on pulmonary assessment is evaluated. The objectives of this study were to describe pulmonary changes as well as autolytic features detected by PMCT examination and to compare those findings with conventional necropsy. Retrospective evaluation of whole-body PMCT images of 46 relatively fresh harbor porpoises and corresponding conventional necropsy reports was carried out, with a special focus on the respiratory tract. Common pulmonary PMCT findings included: moderate (24/46) to severe (19/46) increased pulmonary soft tissue attenuation, severe parasite burden (17/46), bronchial wall thickening (30/46), and mild autolysis (26/46). Compared to conventional necropsy, PMCT more frequently identified pneumothorax (5/46 vs. none), tracheal content (26/46 vs. 7/46), and macroscopic pulmonary mineralization (23/46 vs. 11/46), and provided more information of the distribution of pulmonary changes. These results indicate that PMCT adds information on pulmonary assessment and is a promising complementary technique for necropsy, despite the frequent presence of mild autolytic features.

## 1. Introduction

Wildlife mortality events represent important opportunities in species and ecosystem health assessment; therefore, cetaceans are considered sentinel in monitoring and conservation of the marine ecosystem [1,2,3,4]. A high stranding frequency of harbor porpoises (*Phocoena phocoena*) is documented in countries bordering the North Sea, with hundreds of harbor porpoises washing up on the Dutch coast every year [5,6]. Stranding networks including post-mortem evaluation programs are developed to investigate mortality causes, general health status, and to identify and evaluate the impact of emerging (anthropogenic) threats on the porpoise population and ecosystem health [2,3,7].

As access to imaging facilities increases, application of whole-body cross-sectional imaging techniques in veterinary and wildlife post-mortem programs, otherwise referred to as virtopsy, is advancing [4,8,9,10,11,12]. Therefore, specialized knowledge to interpret these imaging studies arises [4,8,13,14].

Documented advantages of post-mortem cross-sectional imaging techniques in both human and veterinary literature include the non-invasive collection of volumetric datasets of post-mortem findings, with two- and three-dimensional reconstruction possibilities, providing patho-anatomic information with excellent spatial and contrast resolution [15,16]. Additionally, increased operating ease with data exchange for third opinion-seeking purposes and reproducibility, zoonotic risk reduction, and specific guidance during the following conventional necropsy are some of the listed advantages [8,17]. In the field of human forensics, virtopsy in the form of post-mortem computed tomography (PMCT) has proven especially valuable in trauma cases as a fast-imaging tool for identifying and documenting the presence and exact locations of foreign material, gaseous changes, and skeletal alterations [16,17].

PMCT evaluation has successfully been used in the detailed description of anatomical skeletal features of the harbor porpoise vertebral column and skeletal and ear pathology in several cases [18,19,20]. However, pulmonary changes have not been systematically evaluated. Multiple studies report a high incidence of pulmonary pathology in the harbor porpoise [21,22,23,24,25,26]. Literature on cetacean pulmonary PMCT is limited to case reports, with population studies currently lacking [4,11,12,27]. The aim of this study was to compare pulmonary changes detected retrospectively by PMCT in harbor porpoises with the complementary necropsy reports, including comparing the degree and effect of carcass decomposition. It was hypothesized that the effects of carcass decomposition will be visible on PMCT images but will only partially influence pulmonary evaluation, and therefore can become a complementary technique for necropsy.

## 2. Materials and Methods

### 2.1. Data Collection

Carcasses of stranded harbor porpoises in The Netherlands were retrieved between November 2016 and August 2018, in collaboration with the National Stranding Network, and submitted for post-mortem examination conducted at the Faculty of Veterinary Medicine of Utrecht University. Only relatively fresh (decomposition carcass condition (DCC) score 1–3) and complete carcasses were considered eligible and submitted for whole-body CT examination. Pulmonary PMCT findings were not specifically communicated with the pathologist prior to the necropsy, enabling a blinded protocol. Subsequently, conventional necropsy was performed, including histopathological sampling and assessment following international standardized procedures [28].

A whole-body PMCT scan of each porpoise was performed in ventral recumbency by a 64-slice gliding gantry CT scanner (Somatom Definition AS, Siemens AG, München, Germany). Acquisition parameters included: 0.6 mm detector width, 120 kVp, reference 370 mAs, 0.5 s rotation speed, 0.9 pitch, and matrix size 512 × 512. The field of view (FOV) varied in correspondence with the size of the harbor porpoise. Reconstruction parameters included 3 mm slice thickness, 1.5 mm increment, and the soft tissue algorithm (B30f medium smooth) with a window width/window level of 300/50. Additionally, a reconstruction using a bone algorithm (B60F, 1 mm slice thickness, 0.7 mm increment) was performed, that was viewed in two windows to maximize the evaluation of skeletal (window width/level 3000/600) and pulmonary (window width/level 1250/-500) structures.

Macroscopic evaluation of the airways on necropsy was routinely performed in the following order: Position of the diaphragm (assessment of presence of pneumothorax) after opening the peritoneal cavity prior to opening the thoracic cavity. Location and evaluation of tracheal content after opening of the thoracic cavity. Assessment of the pleural cavity (adhesions, fluid) whilst removing heart and lungs. Ex vivo, the lungs were evaluated for asymmetry in size, color, and texture. The presence and severity of parasite burden was graded as previously established by ten Doeschate et al. [26]. Lung lesions were evaluated on the following aspects: size, location, color, texture, shape, margin, and demarcation.

Histopathological sampling of a minimum of two pieces of each lung, one from the hilus (including part of a major bronchial tree) and one near the surface, including pleura, was performed.

### 2.2. Data Evaluation

All PMCT studies were retrospectively evaluated by a veterinary radiology (European College of Veterinary Diagnostic Imaging, ECVDI) resident (N.K.) under the supervision and agreement of a board-certified (ECVDI) diplomate (S.V.). A Digital Imaging and Communication in Medicine (DICOM) format was used to review the images in the previously mentioned window levels, including the use of the multiple planar reconstruction (MPR) tool in the viewer of the Picture Archiving and Communication System (PACS, Enterprise Imaging, Diagnostic Desktop 8.1.2, Agfa Healthcare, Mortsel, Belgium). PMCT images of the lungs were evaluated and scored on the parameters, as described in Table 1.

### 2.3. Carcass Condition

A PMCT scoring system for evaluation of the degree of carcass decomposition associated with lung autolysis was developed with a score ranging from 0 to 3 (Figure 1). This grading was based on several features, including the amount of soft tissue attenuation present in the pulmonary parenchyma, with less than 25% classified as mild, 25–75% as moderate, and over 75% as severe. Presence of retraction of the lung margins with presence of gas in the pleural space in the absence of thoracic wall trauma was considered a feature of moderate autolysis. Gas attenuation present in the vertebral canal and other visible vascular and soft tissues structures was considered a feature of severe autolysis. Presence of pneumothorax was also separately scored as absent or present.

### 2.4. Tracheobronchial Changes

PMCT evaluation of the presence and degree of tracheobronchial intraluminal tubular structures, assessed as an overall average reflecting left- and right-sided lungs, was scored as: none, mild: few parasites present, filling less than 25% of the bronchial lumen, moderate: multiple parasites present, filling 50–75% of the bronchial lumen, and severe: parasites present up to the level of the trachea, almost complete occlusion of bronchial lumen (Figure 2). Presence of tracheobronchial fluid was noted as absent or present. Bronchial wall thickening was scored as absent or present.

### 2.5. Pulmonary Parenchymal Attenuation Changes

Severity categorization of pulmonary soft tissue attenuation (PSTA), assessed as an average reflecting both lungs, was scored as: absence of increased PSTA, mild: minimal focal, heterogeneous PSTA affecting <25% of the lungs, moderate: diffuse, heterogeneous, patchy PSTA, including focal homogeneous PSTA affecting 25–75% of the lung parenchyma, and severe: diffuse, homogeneous PSTA, affecting >75% of the lung parenchyma (Figure 3). Thereafter, distribution of PSTA changes of dorsal–ventral and left–right were noted. Presence of pulmonary parenchymal nodules, mineralization, and air entrapment was evaluated and scored as absent or present.

The necropsy reports were retrospectively assessed and divided into the following parameters presented in Table 2.

## 3. Results

### 3.1. Subjects

A total of 51 harbor porpoises underwent PMCT and conventional necropsy, 5 of which were too autolyzed on histologic evaluation of the lungs to include in the study, leaving 46 cases to analyze. Of these 46 animals, 20 were female and 26 were male, and they were of mixed age (juvenile and adult, no neonates).

### 3.2. Carcass Condition

On PMCT evaluation, all carcasses were considered to have signs of autolysis, with overrepresentation of a mild degree of autolysis (26/46), followed by of a moderate amount of autolysis (16/46). Only 9 out of 46 carcasses were considered to have signs of severe autolysis. Decomposition carcass condition scored during conventional necropsy revealed the majority of the carcasses (24/46) to be within the DCC 2 category, followed by 16/46 in the DCC 1 category and 6/46 in DCC 3.

Pneumothorax was identified in 5 carcasses with PMCT, presenting as uni- or bi-lateral gas attenuation in the pleural cavity and variable lung margin retraction with associated volume loss of the lung lobes. Pneumothorax was, however, not noted upon gross post-mortem assessment.

### 3.3. Tracheobronchial Changes

Tracheal fluid was commonly identified in 26/46 porpoises on PMCT images. Tracheal edema (2/46) and foam (5/46) was less frequently described in necropsy reports.

Tracheobronchial presence of parasites was often visible on PMCT (32/46) and reported during pathologic examination (36/46). Overrepresentation (17/46) of severe parasite burden was noted on PMCT, and a moderate degree most commonly (18/46) observed during necropsy. A mild parasite burden was the least commonly reported in both PMCT (4/46) and gross post-mortem examination (4/46).

Bronchial wall thickening was found on PMCT evaluation in 30/46 animals. All 30 animals presented with tracheobronchial parasites on PMCT, with 17/30 in the severe and 11/30 in the moderate group of PMCT parasite burden. Bronchial wall thickening was not evaluated during gross necropsy examination.

### 3.4. Pulmonary Parenchymal Attenuation Changes

A moderate increase in pulmonary soft tissue attenuation was the most common visible pattern (24/46) on PMCT, followed by a severe PSTA increase (19/46). A mild increase in PSTA was incidentally noted in 3/46 cases. A completely normal presentation of the pulmonary parenchyma was not identified.

Different patterns of increased PSTA distribution were recognized in the moderate and severe groups on PMCT evaluation (Figure 3). This included a pattern of diffuse and symmetrically distributed, inhomogeneous patchy, ill-defined, rounded peribronchiolar areas of ground glass attenuation (Figure 3B). A second pattern recognized could be described as asymmetrically distributed, medium- to large-sized, amorphously shaped, homogeneous soft tissue attenuating areas throughout the pulmonary parenchyma (Figure 3C). A third pattern seen within the severe group was symmetric and homogeneous PSTA of the complete pulmonary parenchyma (Figure 3C).

During PMCT evaluation, PSTA was perceived as asymmetric in 20/46 of the cases, with equal numbers of predominantly left- (10/46) or right-sided (10/46) distribution. Incidentally (2/46), a mainly ventral distribution of increased attenuation was noted, though a primarily dorsal distribution was never seen. In some cases, asymmetric distribution was clearly related to previous positioning of the animal, with flattening of the associated thoracic wall (Figure 4).

In reviewed pathology reports, asymmetry of lung size was infrequently mentioned in 6/46 of the cases. Distribution of lung changes was often not mentioned (30/46), though in 16/46 cases it was described as either bilateral or diffusely present. Pulmonary edema was frequently mentioned (38/46) in either gross necropsy or histology reports.

Nodular pulmonary changes were found in 22/46 animals on PMCT evaluation and mentioned in 11/46 necropsy reports. Pulmonary mineralization was identified in 23/46 animals on PMCT images. Different patterns of mineralization were recognized on PMCT (Figure 5), including singular medium to large (5–15 mm) irregular shaped mineral attenuating areas, often associated with (nodular) consolidated regions (Figure 5A). More often, multifocal distributed pinpoint (±1–5 mm) mineral attenuating foci were visible throughout the lung parenchyma (Figure 5B). During macroscopic assessment, pulmonary mineralization was detected in 11/46 animals. During histologic examination of the lung tissue, mineral deposits, mostly noted in the bronchial walls, were reported in 28/46 of the cases.

PMCT evaluation commonly (33/46) showed pulmonary air entrapment (Figure 6), mostly seen in the periphery of the lung field. Presentation varied from medium to large, thin-walled, peripheral bullous lesions, often with signs of bronchiectasis and bronchial wall thickening (Figure 6A). A second pattern with diffusely distributed small, rounded gas attenuations throughout the pulmonary parenchyma, including intravascular gas attenuation (Figure 6B) commonly in combination with bronchial changes, was identified. Thirdly, a more mosaic pattern with peripheral, varying in size, relatively well-defined areas of decreased pulmonary parenchymal attenuation was observed (Figure 6C).

In 12/46 of the cases, fibrosis was reported, and in 13/46, emphysema, on histologic evaluation of lung tissue.

A summary of the results of PMCT image evaluation and conventional necropsy scores is reported in Table 3.

## 4. Discussion

PMCT presents a valid method for thoracic virtopsy in harbor porpoises. As hypothesized, increased pulmonary soft tissue attenuation and features of decomposition were often visible on PMCT. Despite the presence of these changes, different patterns could still be identified and described. In comparison with conventional necropsy findings, this study indicated the complementary capability of PMCT, in particular to spatially describe and document pulmonary changes, and detect pulmonary mineralization, pneumothorax, and tracheal content.

Thus far, this is the first study to describe PMCT pulmonary findings in retrospect, including PMCT carcass decomposing features, in a larger number of harbor porpoises. Despite commonly reported pulmonary disease [22,23,24,25,26,29], only few studies have described PMCT cetacean pulmonary imaging features [4,11,12,27], which are mainly singular case reports of different delphinid species.

Both carcass condition scores revealed grossly similar subcategorization of carcass condition with overrepresentation of the group with mild signs of autolysis on PMCT (26/46) and the relatively fresh condition (DCC2) on gross pathology evaluation (24/46). Considering the mild decomposition scores and the presence of increased lung attenuation in all cases, it is likely that internal pulmonary livores (hemoconcentration and congestion in position-dependent lung vasculature and parenchyma) is also one of the first post-mortem changes to develop in the harbor porpoise, as stated by Levy et al. [30] in human PMCT studies. This statement is further supported by the frequently identified presence (22/46) of a gradient distribution of increased pulmonary attenuation on PMCT evaluation, sometimes clearly related to the (previous) position of the carcass, as revealed by unilateral flattening of the thoracic wall (Figure 4). It needs to be kept in mind that stranded harbor porpoises are mostly found in lateral recumbency, and therefore a lateralized left or right distribution is more commonly found than a dorsal or ventral distribution.

A varying degree of pneumothorax was identified on PMCT evaluation in 5/46 carcasses, that was not reported on conventional necropsy, likely due to the escape of gas the moment the thoracic cavity is opened. This proposes PMCT examination to be superior to conventional necropsy in detecting pneumothorax as has been documented in human literature by Thali et al. [17].

A more frequent identification of tracheal content was seen on PMCT evaluation in 26/46 animals, compared to 7/46 animals in conventional necropsy, and could potentially add more information in the process of post-mortem examination. The increased detection rate of tracheal content on PMCT could be explained by gradual dissolvement and efflux of tracheal foam and fluid, partly due to change of position and with time passing between the PMCT and necropsy. These would be interesting parameters to include in a future study design. Tracheobronchial fluid has often been documented in human literature in non-drowning cases and is thought to be formed in the process of decomposition [31]. However, inconsistent reporting of tracheobronchial foam and fluid aspiration as a sign of underwater entrapment and drowning can be found in both human and veterinary forensic literature [32,33]. As in cetacean species, a so-called ‘dry’ asphyxia-induced way of drowning (with vagal-induced laryngospasm, cardiac arrest, and ultimately, cerebral hypoxia leading to death) is reported [29,34,35,36], and the tracheal fluid or foam content may be present secondary to pulmonary edema.

Bronchial wall thickening (30/46) was one of the most reported findings on PMCT evaluation in the current study and was always seen in combination with pulmonary helminthiasis. Tracheobronchial helminths would likely act as an irritant on the respiratory epithelium, initiating an inflammatory reaction and mineralization of the wall tissue layers in a chronic phase, resulting in the visualized thickening of the walls. As bronchial wall thickness is more difficult to assess on conventional necropsy, this represents an important, potentially more quantitative evaluation of the lungs on PMCT imaging. Another potential sign of bronchial disease in this study was thought to be represented by the high presence of pulmonary air entrapment on PMCT images (33/46). These hypoattenuating pulmonary changes represented in different patterns (Figure 6), with medium to large, mainly peripherally located, thin-walled bullous-like lesions in the direction of the bronchi (Figure 6A), likely representing what has been described as saccular bronchiectasis [37]. In theory, pulmonary air entrapment, including intravascular gas attenuations, could also represent putrefaction gases [16,30,38], bullous emphysematous changes (also called honeycombing pattern), secondary chronic obstructive and/or fibrotic pulmonary disease [37,39], subpleural emphysema, or decompression of supersaturated tissues secondary to the process of drowning/asphyxia at depth [27,32,36].

In this study, all subjects showed pulmonary changes on PMCT imaging, with the most common findings being a moderately (24/46) and severely (19/46) increased PSTA. Main CT differentials of increased PSTA include bronchopneumonia, contusion/hemorrhage, atelectasis, edema, and in case of PMCT imaging, also post-mortem hypostasis leading into edema [30,37,40,41]. Pulmonary edema was one of the most reported findings on conventional necropsy (38/46). Differentiating lung pathology with PMCT by means of pulmonary patterns has not been described in harbor porpoises and only infrequently in other cetacean species [4,27]. Several studies in the field of human forensic imaging identified distinct PMCT pulmonary patterns and have tried to correlate these to specific pulmonary pathology confirmed with conventional autopsy [16,17,38,40,41,42,43,44,45,46]. These studies repeatedly reported autopsy-confirmed PMCT ‘drowning’ lung changes represented by a mosaic pattern of centrolobular distributed, patchy to nodular ground glass opacities, although no consistent correlation between PMCT lung patterns and autopsy-confirmed drowning could be made. In the current study, a similar centrolobular patchy PSTA pattern has been identified in several harbor porpoises (Figure 3B) and could be a potential additional criterium in an equivalent diagnosis of drowning or bycaught [29,32,36]. Nonetheless, as no consistent relation is established in the literature, evaluation of the complete constellation of features using both conventional necropsy and PMCT remains important to lead to an ultimate diagnosis. The main challenge in pulmonary PMCT imaging interpretation is to differentiate non-specific post-mortem hypostasis (known as internal livores) and distinguish these from specific signs that may represent pathological changes [16,38,41]. Future prospective design studies could be helpful in attempting to establish a more consistent relation between specific PMCT lung patterns and ultimate diagnosis on pathology.

Pulmonary nodular changes were more commonly reported on PMCT evaluation (22/46), compared to necropsy (11/46), as well as an increased visualization of pulmonary mineralization on PMCT (23/46), compared to macroscopic evaluation during necropsy (11/46). This is likely related to the increased spatial evaluating abilities of PMCT of the lungs in situ, resulting in a more specific lesion distribution description, which then can be followed by guided sampling methods during necropsy. However, PMCT evaluation of pulmonary nodular changes in the current study was sometimes found to be hampered in cases with a severe, diffuse increase in pulmonary attenuation, and represents a pitfall of PMCT pulmonary evaluation of severely affected or decomposed lungs (Figure 3D). PMCT scanning with an endotracheal tube in place and maintaining a constant positive-pressure ventilation during acquisition could potentially partially overcome this problem in more fresh cadavers [4].

## 5. Conclusions

In conclusion, this paper described the advantages of pulmonary PMCT in 46 harbor porpoises. Due to its cross-sectional nature and reconstruction possibilities, an overall higher detection rate of pneumothorax, tracheal fluid, macroscopic pulmonary mineralization, severity of helminth infestation, and characterization of bronchial changes was found on PMCT evaluation. Based on these findings, application of PMCT in harbor porpoises is considered a valuable asset, complementary to necropsy. Incorporation into a standardized post-mortem examination protocol is recommended. The small size and low bodyweight of the harbor porpoise in particular makes PMCT a relatively uncomplicated and accessible post-mortem examination method to perform, compared to other larger and heavier cetacean species. Despite the overall high degree of pulmonary soft tissue attenuating changes and decomposing features present, several imaging lung patterns were found, similar to pulmonary patterns described in human forensic cross-sectional imaging literature. It is important to emphasize that, in line with previous literature, none of these described patterns were found pathognomic for certain pathology. Potential further improvement of pulmonary evaluation on PMCT could be reached by positive-pressure ventilation during image acquisition, with further quantification of parameters such as bronchial wall thickness. Imaging relatively fresh cadavers would be highly recommended as results in this study showed mild to moderate signs of autolysis in an early phase, despite mild external decomposition features (low DCC scores). Access to an onsite CT scanner would be optimal to prevent development of early signs of autolysis. Lungs of frozen/thawed specimens would therefore likely display too many autolytic artefacts on PMCT to appropriately evaluate the pulmonary parenchyma. As the current study had a retrospective design, prospective studies using pulmonary positive-pressure ventilation PMCT in combination with conventional necropsy and (guided) tissue sampling are needed to further characterize lung changes in the harbor porpoise species.

## Figures and Tables

**Figure 1 animals-12-01454-f001:**
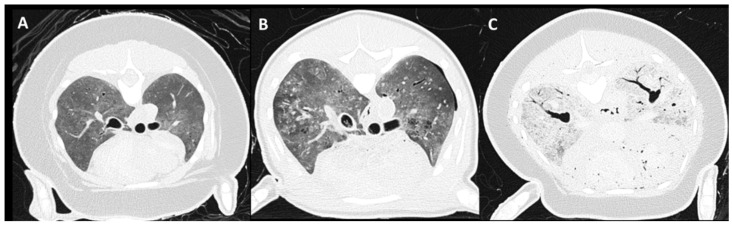
Examples of autolysis score on PMCT. From left to right: (**A**) mild: PSTA (pulmonary soft tissue attenuation) present in less than 25% of the lung parenchyma, (**B**) moderate: retraction of the lung margins with presence of gas in the pleural space and 25–70% of PSTA noted, and (**C**) severe: gas attenuation presence in vascular structures and other visible soft tissue structures, over 75% of PSTA present.

**Figure 2 animals-12-01454-f002:**
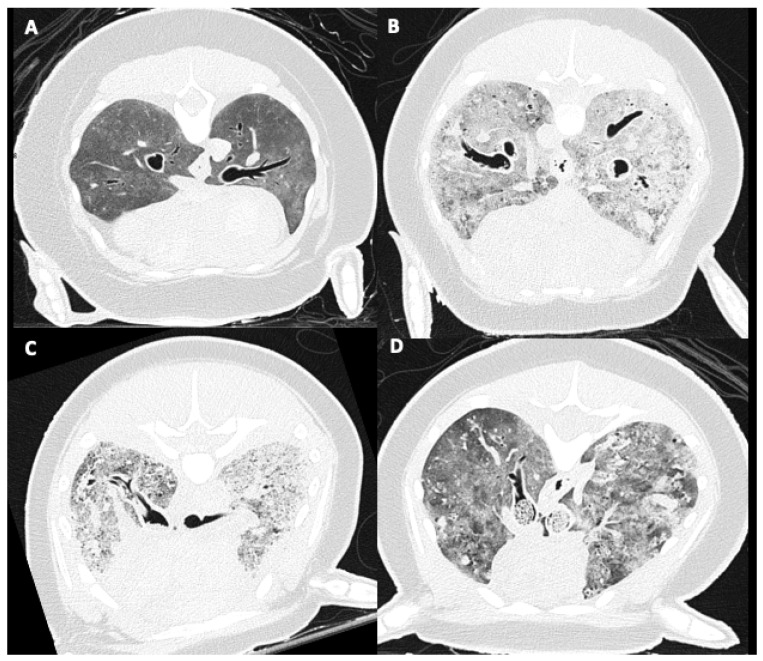
Intraluminal tracheobronchial tubular structures’ severity score on PMCT. (**A**) Absent, (**B**) mild: few parasites present, filling less than 25% of the bronchial lumen, (**C**) moderate: filling 50–75% of the bronchial lumen, and (**D**) severe: parasites present up to the level of the trachea, almost complete occlusion of bronchial lumen.

**Figure 3 animals-12-01454-f003:**
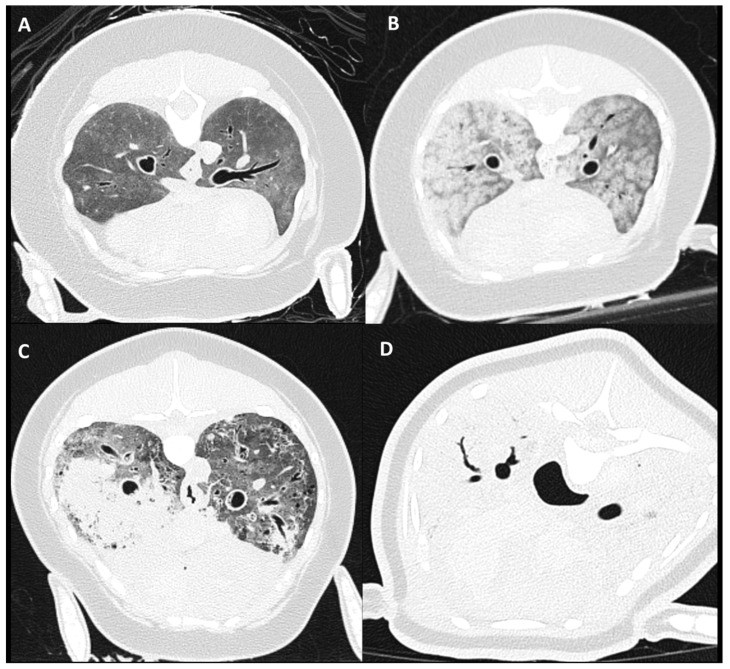
Pulmonary soft tissue attenuation score on PMCT. Clockwise presentation of: (**A**) mild: minimal focal, heterogeneous PSTA affecting <30% of the lungs, (**B**,**C**) moderate: diffuse, heterogeneous PSTA, including focal homogeneous PSTA affecting 30–75% of the lung parenchyma, and (**D**) severe: diffuse, homogeneous PSTA, affecting >75% of the lung parenchyma.

**Figure 4 animals-12-01454-f004:**
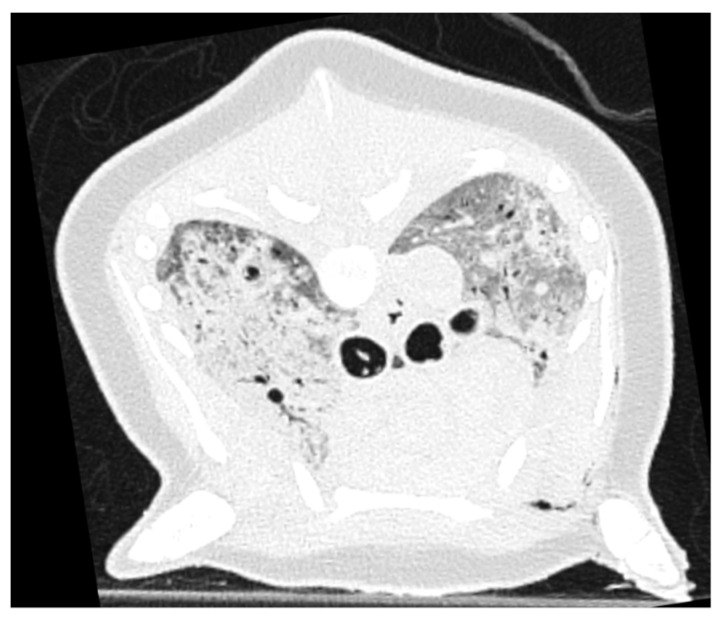
Example of an asymmetric gradient of pulmonary soft tissue attenuation found on PMCT, with signs of associated thoracic wall flattening on the left side of the image, likely consistent with positional-dependent hypostatic edema.

**Figure 5 animals-12-01454-f005:**
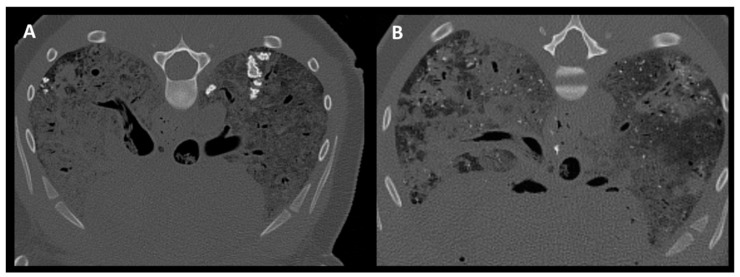
Two PMCT patterns of pulmonary mineralization. (**A**) Singular medium to large irregular shaped mineral attenuating areas, associated with a (nodular) consolidated region. (**B**) Multifocal distributed pinpoint mineral attenuating foci throughout the lung parenchyma.

**Figure 6 animals-12-01454-f006:**
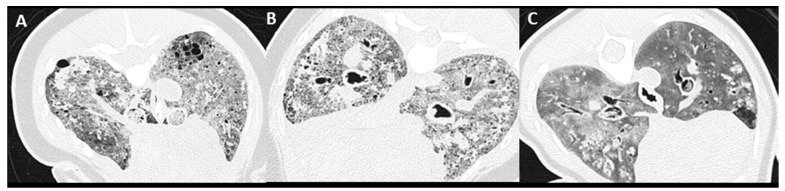
Patterns of pulmonary air entrapment on PMCT. From left to right: (**A**) Peripheral bullous bronchiectasis, (**B**) diffusely distributed small, rounded gas attenuations throughout the pulmonary parenchyma, including the vasculature, and (**C**) mosaic pattern with peripheral areas of decreased parenchymal attenuation.

**Table 1 animals-12-01454-t001:** PMCT pulmonary evaluation.

PMCT Parameter	Score
PMCT autolysis score	0 absent, 1 mild, 2 moderate, 3 severe
Pneumothorax	0 absent, 1 present
**Tracheobronchial**	
Intraluminal tracheobronchial tubular structures	0 none, 1 mild, 2 moderate, 3 severe
Bronchial wall thickening	0 absent, 1 present
Tracheobronchial fluid	0 absent, 1 present
**Pulmonary attenuation**	
Severity of pulmonary soft tissue attenuation	0 absent, 1 mild, 2 moderate, 3 severe
Dorsoventral distribution pulmonary soft tissue attenuation	0 diffuse, 1 dorsal, 2 ventral
Left/right distribution pulmonary soft tissue attenuation	0 bilateral, 1 left, 2 right
Pulmonary parenchymal nodules	0 absent, 1 present
Pulmonary mineralization	0 absent, 1 present
Pulmonary air entrapment	0 absent, 1 present

**Table 2 animals-12-01454-t002:** Conventional necropsy report evaluation.

Parameter	Score
Decomposition carcass condition (DCC)	1–3
Pleural space content	0 absent, 1 present
**Tracheobronchial**	
Tracheal content	0 absent, 1 edema, 2 foam
Tracheobronchial helminths	0 none, 1 mild, 2 moderate, 3 severe
**Pulmonary parenchyma**	
Pulmonary edema	0 absent, 1 present
Asymmetry lung volume	0 absent, 1 present
Distribution pattern pulmonary lesions	0 not mentioned, 1 diffuse, 2 cranioventral, 3 mid-dorsal
Pulmonary nodules	0 absent, 1 present
Pulmonary mineralization macroscopically	0 absent, 1 present
Pulmonary mineralization microscopically	0 absent, 1 present
Histologic tissue reactions	0 none mentioned, 1 fibrosis, 2 emphysema

**Table 3 animals-12-01454-t003:** PMCT and conventional necropsy evaluation score results.

PMCT	Number (%)	Conventional Necropsy	Number (%)
**Autolysis Score**		* **DCC** *	
None	0/46 (0)		-
Mild	26/46 (57)	1	16/46 (35)
Moderate	16/46 (35)	2	24/46 (52)
Severe	4/46 (9)	3	6/46 (13)
Pneumothorax	5/46 (11)	Pleural content	0/46 (0)
**Tracheobronchial changes**			
Tracheobronchial tubular structures		Helminthiasis	
Absent	14/46 (30)	Absent	10/46 (22)
Mild	4/46 (9)	Mild	4/46 (9)
Moderate	11/46 (24)	Moderate	18/46 (40)
Severe	17/46 (37)	Severe	14/46 (30)
Bronchial wall thickening	30/46 (65)		-
Tracheobronchial content		Tracheal content	
None	20/46 (43)	None	39/46 (85)
Fluid	26/46 (57)	Edema	2/46(4)
		Foam	5/46 (11)
**Pulmonary parenchyma**			
PSTA severity		Pulmonary edema	38/46 (83)
None	0/46 (0)		-
Mild	3/46 (7)		-
Moderate	24/46 (52)		-
Severe	19/46 (41)		-
		Asymmetry lung volume	6/46 (13)
**Distribution lung changes**			
Dorsal	0/46 (0)		-
Ventral	2/46 (4)		-
Left	10/46 (22)		-
Right	10/46 (22)		-
Bilateral	26/46 (56)	Bilateral	16/46 (35)
Parenchymal nodules	22/46 (48)	Parenchymal nodules	11/46 (24)
Mineralization	23/46 (50)	Gross mineralization	11/46 (24)
Pulmonary air entrapment	33/46 (72)	-	-
**Histology**			
	-	Histologic mineralization	28/46 (61)
	-	Histologic emphysema	13/46 (28)
	-	Histologic fibrosis	12/46 (26)

## Data Availability

Data is contained within the article.

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
