# Peer review of "Post-Mortem Computed Tomography Pulmonary Findings in Harbor Porpoises (Phocoena phocoena)"

_animals, 2022, doi:10.3390/ani12111454_

Round 1

Reviewer 1 Report

Review: animals-1731765

Title: Post Mortem Computed Tomography Pulmonary Findings in Harbor Porpoises

Summary

·                First off I would like to state that I reviewed this as a clinical radiologist with expertise in human post-mortem imaging and not as a veterinary radiologist. So I cannot comment on specific pathological findings in harbor porpoises.

·                It is good to see that post-mortem CT (PMCT) is also starting to be used in veterinary radiology. In human radiology PMCT has shown to be a great asset as an ancillary diagnostic tool next to the autopsy and in some cases even replacing the autopsy.

·                The manuscript is well written and a pleasure to read.

Specific comments

Abstract

·                No comments.

Introduction

·                No comments and a clearly stated study aim is presented.

Materials and methods

·                Page3, line 84: Were the pathologists aware of the PMCT findings?

·                Page 4, line 134: Do the authors have information of the time between discovery of the harbor porpoises and the time of the PMCT? This would perhaps give some insight in the amount of putrefaction that could be expected. In humans intravascular gas formation can be found within hours after demise. In humans a radiological alteration index has been proposed but in larger validation studies this didn’t work well (reference: Egger C, Vaucher P, Doenz F, Palmiere C, Mangin P, Grabherr S. Development and validation of a postmortem radiological alteration index: the RA-Index. Int J Legal Med. 2012 Jul;126(4):559-66). As a result most radiologists and pathologists will not make a statement about the time of death based on PMCT.

Results

·                Page 7, line 210: In humans tracheal fluid is a normal finding. This is discussed in the discussion, but I wonder if there is any pathology literature on this finding in harbor porpoises. It could well be that during an autopsy this is difficult to detect let alone score.

·                Page 7, line 234: Was the asymmetry in keeping with the position in which the harbor porpoise was found (or even scanned)?

Discussion

·                Well written and the authors present the pro’s and con’s of PMCT.

·                Page 12, line 368: Although indeed PMCT will give a better overview of the lungs it is also known that the diagnostic sensitivity is lower. Latten et al. performed a study in which they compared PMCT to an autopsy. They found that ‘Of the 60 included cases 44% of the CT-guided postmortem biopsies in the left lung and 30% in the right lung showed false negative findings, primarily concerning a bronchopneumonia’. (reference: Latten BGH, Bakers FCH, Hofman PAM, Zur Hausen A, Kubat B. The needle in the haystack: Histology of post-mortem computed tomography guided biopsies versus autopsy derived tissue. Forensic Sci Int. 2019 Sep;302:109882).

·                Page 12, last paragraph: I think that the authors could be more vocal about the application of PMCT in veterinary medicine. They’ve shown it to be of additional value, if an veterinary autopsy facility has access to a CT scanner they should strongly consider to perform PMCT in all cases prior to autopsy. This is in many human (forensic) pathology facilities the standard way of practice.

References

·                No comments.

Tables

·                Table 3: It would be interesting to see where PMCT and autopsy findings overlapped and where they differed. For some categories this won’t be possible but where possible this could be presented.

Figures

·                The illustrations are of good quality and add value to the manuscript.

Author Response

Dear reviewer

Thank you very much for taking the time and effort to assess this manuscript. Also, many thanks for the kind words regarding the general writing style of the manuscript. I would like to reply to your comments and suggestion one by one in the section below (see attached Word file below).

Most kind regards 

Nienke Kuijpers

Reviewer 2 Report

Thank you for the opportunity to review this manuscript. It is always a pleasure to read about advancements in marine mammal science. I have provided in text comments / line by line edits in the PDF attached. Overall I think this is a well conducted study with results that contribute to advancing marine mammal necropsy techniques and understanding of both post mortem changes and the pathophysiology of lung disease in stranded harbour porpoises. I have made some minor suggestions throughout the manuscript. I am curious as to whether you have additional gross necropsy colour images - with this being published online which could contribute to the comparison of CT images with gross findings. 

Author Response

Dear reviewer

Thank you for taking the time and effort to review this manuscript. I am very glad to read you consider this study as a contribution for marine mammal necropsy techniques. Your opinion and remarks on this paper will be taken into serious consideration. According to gross necropsy color images, I have looked through the available case based photographs. Unfortunately, no representable photo image of specific lung lesions seen with PMCT is available, mainly due to the retrospective nature of this study with a specific focus on the respiratory tract. We will take this into consideration for future study design, as it is our intention to more fully compare PMCT findings with necropsy.

I will go through your remaining remarks one by one in the sections below (see attached Word file).

Most kind regards

Nienke Kuijpers
